# Community-Level Participation in Volunteer Groups and Individual Depressive Symptoms in Japanese Older People: A Three-Year Longitudinal Multilevel Analysis Using JAGES Data

**DOI:** 10.3390/ijerph18147502

**Published:** 2021-07-14

**Authors:** Motoki Tamura, Shinji Hattori, Taishi Tsuji, Katsunori Kondo, Masamichi Hanazato, Kanami Tsuno, Hiroyuki Sakamaki

**Affiliations:** 1School of Health Innovation, Kanagawa University of Human Services, Kawasaki 210-0821, Japan; ktsuno-tky@umin.ac.jp (K.T.); hiroyuki.sakamaki@nifty.com (H.S.); 2Research Department, Institute for Health Economics and Policy, Tokyo 105-0003, Japan; shin-ji.hattori@ihep.jp; 3Center for Preventive Medical Sciences, Chiba University, Chiba 263-8522, Japan; tsu-ji.taishi.gn@u.tsukuba.ac.jp (T.T.); kkondo@kkondo.net (K.K.); hanazato@chiba-u.jp (M.H.); 4Faculty of Health and Sport Sciences, University of Tsukuba, Tokyo 112-0012, Japan; 5Center for Gerontology and Social Science, National Center for Geriatrics and Gerontology, Obu 474-8511, Japan

**Keywords:** social capital, contextual effects, depression risk, social contagion

## Abstract

Background: The current study aimed to investigate the contextual effect of volunteer group participation on subsequent depressive symptoms in older people. Methods: We analyzed the longitudinal data of 37,552 people aged 65 years and older in 24 municipalities surveyed in the Japan Gerontological Evaluation Study. Volunteer group participation of older people was assessed in 2013 by one question and depressive symptoms were assessed by the Geriatric Depression Scale 15 in 2016. To investigate a contextual effect, we aggregated individual-level volunteer group participation by each residence area as a community-level independent variable. We conducted a two-level multilevel Poisson regression analysis using the Random Intercepts and Fixed Slopes Model. Results: The average proportion of community-level volunteer group participation was 10.6%. The results of the Poisson regression analysis showed that community-level volunteer group participation reduced the risk for the onset of depressive symptoms by 13% with a 10 percentage point increase in participation, after adjusting for sex, age, population density, total annual sunshine hours and annual rainfall (incident rate ratio, 0.87; 95% confidence interval, 0.78–0.98). Conclusions: Older people living in areas with higher volunteer group participation had a lower risk of developing depressive symptoms regardless of whether or not they participated in a volunteer group.

## 1. Introduction

Depression is an important issue of public health in the world [1]. In particular, the prevalence of depression among aged 65 and older people is estimated to be between 5 and 10% and is projected to be the second leading cause of disease burden in this population by the year 2020 [2,3]. Additionally, a few studies have suggested that depressive symptoms in older people are strongly associated with being housebound and isolated, which may lead to a decline in physical and cognitive functions and eventually to premature death [4]. Depression in older people is one of the risk factors for ameliorable dementia and persistent depression may lead to the development of dementia [5]. The proportion of people over 65 years of age in the total population, i.e., the aging rate, continues to increase, especially in developed countries. In Japan, the proportion of older people in the total population is estimated to increase from 26.6% in 2015 to 38.1% in 2060 [6]. Thus, the prevention of depressive symptoms in the increasing number of older people is becoming an important public health issue to reduce the risk of developing dementia or premature death.

Volunteer activities may have a potential protective factor for depression among older people. For instance, a four-year follow-up study of 6000 Japanese individuals aged 55 to 64 years reported that individual volunteer activities were significantly associated with a reduced risk of developing depression [7]. Volunteer activities are defined as “spontaneous activities that voluntarily contribute to others and society” [8] and previous studies have reported that volunteer activities among older people are associated with psychological health such as life satisfaction, depression, a sense of self-control and a healthy self-esteem [9,10]. As a social mechanism, it is also hypothesized that volunteer activities can expand human relationships and provide new social support and social networks [11,12].

Recent studies have shown that people in areas with a greater social participation have higher levels of health even if they are not in social participation because of the rich social capital [13,14,15]. This characteristic of the community’s influence on an individual’s health is called the contextual effect [16]. In other words, the higher the social capital at the community level, the better the health of the individual possible. Community-level social participation has been strongly or moderately associated with the subjective health perceptions and health indicators of older people, such as depressive symptoms, even after adjusting for individual-level social participation [17]. Another previous study suggested that individual participation mediated the association between community-level social capital and the onset of depressive symptoms [18]. Previous studies have reported the differences in the effects of the types of social participation [19] and individual volunteer activities did not reduce depressive symptoms in individuals [19]. However, it is not clear whether the areas with a high proportion of volunteer group participation have higher levels of individual health or not, regardless of their individual participation in volunteer groups.

The present study aimed to investigate whether older residents living in community areas with higher volunteer group participation of older people are less likely to have subsequent depressive symptoms compared with those living in community areas with a lower prevalence of such participation after adjusting for individual-level volunteer group participation. It was hypothesized that the proportion of community-level volunteer group participation was negatively associated with the onset of depressive symptoms.

## 2. Materials and Methods

### 2.1. Study Design and Subjects

We used panel data from the Japan Gerontological Evaluation Study (JAGES) [20]. The JAGES is an ongoing cohort study investigating social and behavioral factors related to the loss of health concerning a functional decline or cognitive impairment among individuals 65 years old or older. This cohort consists of physically and cognitively independent individuals who have not received any required support and have not needed long-term care certification under the Japanese long-term care insurance system.

The baseline survey was conducted in December 2013 via self-reported questionnaires distributed by mail to 154,496 people 65 years or older selected from 24 municipalities in Japan. Among those who answered the baseline questionnaire (*N* = 103,664, response rate, 67.1%), 1192 were excluded because their residential area was unknown. In order to obtain a stable count for calculating community-level volunteer group participation proportions, we then excluded 647 people living in the community areas with fewer than 30 respondents per area [21]. Thus, 101,825 people were invited to answer the follow-up survey in 2016. In the present study, one community area is equivalent to an elementary school district or a junior high school district, which is a range of the community area that the older people in Japan can access on foot or by bicycle [22].

Participation in the JAGES project was voluntary and those who agreed to participate in the survey returned the survey questionnaire. The research plan was approved by the Research Ethics Committee of the Graduate School of Health Innovation, Kanagawa University of Human Services (approval number: SHI-19–006).

### 2.2. Measurements

#### 2.2.1. Individual-Level Depressive Symptoms

Depression symptoms were measured by the 15-item Geriatric Depression Scale (GDS-15) [23,24] both at the baseline and follow-up. GDS-15 assesses whether respondents have experienced depressive symptoms in their daily lives and a higher score equates to having greater depressive symptoms. Response options were “yes” and “no” and those who had ≥ 8 responses to GDS items both in the baseline and follow-up survey were included in the analyses. Depressive symptoms are defined as having 5 points or higher of GDS-15 [25,26]. GDS-15 has been used in previous studies [23,24,25,27] to evaluate the risk of developing mild or severe depression and good levels of validity and reliability have been reported. For instance, Cronbach’s α coefficient for the internal consistency of the scale was 0.80 [23] and the overall sensitivity and specificity were found to be 0.97 and 0.95, respectively [23].

#### 2.2.2. Individual- and Community-Level Participation in Volunteer Groups

Participants were queried on their frequency of volunteer group participation with one question at the baseline survey in 2013. Response options were ≥ 4 days/week, 2–3 days/week, 1 day/week, 1–3day(s)/month, a few days/year or zero. In the present study, participating in volunteer group activities was defined as participating in more than 1 day/month [28]. In previous studies, volunteer activity of more than 1 day/month was found to be effective in maintaining daily functioning [28] but there is no consensus on the optimal cutoff point.

At the individual level, individual participation in volunteer groups was used as an independent variable. Community-level participation in volunteer groups was calculated by averaging individual volunteer group participation in each community area and the proportion was tabulated as an independent variable for the community areas. The proportion of community volunteer group participation was centralized and then multiplied by 10% so that an estimate per 10 percentage points for participation was calculated by feeding the value into the model. Additionally, cross-level interaction terms were created by multiplying the value centering on individual volunteer group participation by the proportion of community volunteer group participation. As a result, three independent variables were created: (1) the individual volunteer group participation, (2) the proportion of community volunteer group participation that was estimated at 10% and (3) the cross-level interaction terms.

#### 2.2.3. Individual-Level Covariates

Individual-level covariates were sex (men and women), age (65–69, 70–74, 75–79, 80–84 and ≥ 85 years old), education (less than 6 years, more than 7 years and missing), equivalent income (JPY < 2,000,000, JPY 2,000,000–3,999,999, JPY ≥ 4,000,000 and missing), living alone (living with others, living alone and missing), drinking alcohol (never, past, current and missing), smoking (never, past, current and missing), chronic disease (yes and no) and body mass index (BMI (kg/m^2^) < 18.5, ≥ 18.5 and missing).

Having chronic disease was measured by asking participants if they were currently receiving any medical treatment related to strokes, hypertension, diabetes or ear disease; response options were “yes” and “no”. These four diseases of missing values were set as MNAR (Missing Not At Random) and were included in the “no” category. BMI (kg/m^2^) was calculated using the height and weight that the participants answered in the questionnaire. Regarding BMI (kg/m^2^), previous studies suggest that there is no consensus on the association between weight and depressive symptoms [29,30]. Based on this, three categories of BMI (kg/m^2^) were used in the current study: thin(< 18.5), others (≥ 18.5) and missing [29]. If participants did not respond to the individual-level covariates, corresponding observations were assigned to “missing” categories.

#### 2.2.4. Community-Level Covariates

Population density, sunshine hours and rainfall were used as covariates at the community level because these variables are associated with volunteer activities or depressive symptoms [31,32,33]. For example, the volunteer participation rate rapidly declines when the population density of residential areas exceeds 4000 people per square kilometer [31]. In other words, the lower the population density of inhabitable areas is, the more active volunteer participation the residents engage in is. The population density per square kilometer (persons/km^2^) of the 522 community areas were in quartiles calculated and classified into four category variables (< 5066, 5092–8929, 8968–11,282 and ≥ 11,300 persons/km^2^).

Previous research has suggested that weather changes such as sunshine and rainfall are associated with mood disorders [32]. In the same manner, annual total hours of sunshine for the 522 community areas were in quartiles calculated and classified into four category variables (< 1912, 1912–2047, 2047–2075 and ≥ 2076 mean hours). The Japan Meteorological Agency (JMA) has a definition for the sunshine duration. According to the JMA’s website, the duration of sunshine is defined as the time when direct sunlight irradiates the ground surface and “direct irradiance is 0.12 kW/m2 or more” [34]. Annual rainfall (mm) for the 522 community areas were in quartiles calculated and categorized into four category variables (< 1483, 1483–1520, 1521–1621 and ≥ 1622 mm). Rainfall is defined by the Ministry of Land, Infrastructure, Transport, and Tourism (MLIT). According to the website of the MLIT, “(t)he amount of rainfall is the depth of water if the rainfall does not run off anywhere and accumulates as it is,” and “(t)he amount of rainfall is the risk of rainwater not running off anywhere else, not evaporating, and not seeping into the ground and so on, and how deep it goes” [35].

### 2.3. Statistical Analysis

To examine the contextual effects of volunteer group participation on the risk of individual subsequent depressive symptoms, we conducted a two-level multilevel Poisson regression analysis (level 1 for individuals and level 2 for communities) using the Random Intercepts and Fixed Slopes Model. The incidence rate ratio (IRR) and 95% confidence interval (CI) were calculated. The reason for using the Poisson regression analysis was because the prevalence of depressive symptoms (10.4%) in the current study was greater than 10% [36]. Three models were created for the analysis. In Model 1, depressive symptoms as the dependent variable, individual-level volunteer group participation, the proportion of community-level volunteer group participation and cross-level interaction terms as independent variables were entered, adjusted for age and sex. In Model 2, all community-level covariates were additionally added. In Model 3, other individual-level covariates were additionally added. IRRs and 95% CIs had calculated estimates per 10 percentage points of community volunteer group participation. All statistical analyses were performed using STATA 16/SE (Stata Corp LLC, College Station, TX, USA).

## 3. Results

Figure 1 shows a flowchart of the participant inclusion and exclusion criteria. Among the 101,825 baseline eligible respondents, 61,267 answered the follow-up questionnaire in December 2016 (follow-up rate: 59.8%). The number of residential areas was 792. Those who had not died, not moved or not received any required long-term care certification and were available for 3 years of follow-up were eligible for our study. Among them, 9231 people with a missing response for participation in volunteer activities in 2013, 3138 people with GDS-15 answers of ≤ 8 items (i.e., ≥ half of the items were missing) and 255 people who had depression, dementia and Parkinson’s disease in 2013 were excluded. As our dependent variable was the onset of depressive symptoms, we also excluded people who had depressive symptoms at the baseline (*n* = 9070). The final analytic participants were 37,522 people (17,575 males and 19,977 females, mean age 72.6, mean (SD) 5.4 years old) nested in 522 community areas.

Table 1 shows the descriptive statistics for each variable at the individual level. In 2013, 5830 (7.5%) participated in a volunteer group 1 day/month or more often. In 2016, 3908 (10.4%) newly-developed depressive symptoms. At the community level, participation in volunteer groups ranged from 1.1%–27.4%. For individual-level covariates, a higher prevalence of depressive symptoms (≥ 10.4%) was observed in people who did not participate in the volunteer group, women, ≥ 75 years old, < 9 years of education, people with an equivalent income of < JPY 2 million, those currently receiving any medical treatment, those living alone, past drinkers, non-drinkers, current smokers and BMI (kg/m^2^) < 18.5 or ≥ 25.

Table 2 shows the IRRs and 95% CIs calculated by the multilevel Poisson regression analysis. In Model 1, individual volunteer participation (IRR, 0.70; 95% CI, 0.63–0.78) as well as the proportion of community volunteer group participation (IRR, 0.86; 95% CI, 0.77–0.97) was statistically significant in lowering the risk of developing depressive symptoms. In Model 2, after additionally adjusting for population density, annual sunshine hours and annual rainfall, this association remained significant. In Model 3, after additionally adjusting for individual covariates, individual volunteer participation (IRR, 0.72; 95% CI, 0.65–0.80) was statistically significant in lowering the risk of developing depressive symptoms whereas the proportion of community volunteer group participation (IRR, 0.90; 95% CI, 0.80–1.01) was not statistically significant in lowering the risk of developing depressive symptoms. A greater age was statistically significant with the risk of developing depressive symptoms in either model. The cross-level interaction term was not statistically significant with a lower risk of developing depressive symptoms in either model.

## 4. Discussion

In the present study, we investigated the association between the proportion of community-level volunteer group participation and an individual’s risk of developing depressive symptoms. In the results, a higher proportion of community-level volunteer group participation was found to lower the risk of developing depressive symptoms among all older people, regardless of whether or not he/she participated in a volunteer group. To the best of our knowledge, this is the first study that has shown the contextual effect of volunteer group participation of older people on lowering the risk of developing depressive symptoms. Our findings, which showed that community-level volunteer group participation reduces the risk of developing depressive symptoms, support previous studies reporting that community-level social participation reduces the risk of developing depressive symptoms [17].

Social contagion, one of the components of social capital, is a potential pathway that explains the contextual effects of community-level volunteer participation on individual-level health [37]. Social contagion refers to the notion that behaviors spread more quickly through a tightly-knit social network [38]. Social contagion may prevent depressive symptoms in community areas where community-level volunteer group participation is more prevalent by increasing the frequency of outings through volunteer activities of older people and increasing human contact opportunities. In addition, being proactive in volunteer activities in the community areas may increase the chances of receiving social support from others. Increased opportunities to receive social support from others may be associated with a lower risk of developing depressive symptoms even if he/she does not participate in volunteer activities. This is also in line with the studies reporting that social support is associated with a lower risk of developing depressive symptoms [39,40,41].

The results of the present study also revealed that individuals who participated in a volunteer activity at least once a month had a 28% lower risk of developing depressive symptoms. The possible reasons why individual participation in volunteer activity prevents the onset of depressive symptoms are because of the increase in physical activity due to volunteer group participation [42], the increase in social networks and social support [39,40,41] and the acquisition of a sense of fulfillment and purpose in life through obtaining a role [43].

In the present study, cross-level interactions were not statistically significant in any model. In other words, the expected lower risk of developing depressive symptoms associated with a higher proportion of community-level volunteer group participation was not different between individual-level volunteer group participation and non-participation, suggesting the effect was homogeneous.

The results of the study also suggest the risk of developing depressive symptoms increases with aging. The risk of developing depressive symptoms in older people is known to be influenced by chronic diseases such as hypertension and diabetes. This result is indicative of previous studies supporting a higher risk of developing depressive symptoms in older people [4]. As the results in this study showed that the risk of developing depressive symptoms was 8% higher in the age group 70–74 years compared with 65–69 years, if volunteer group participation increased by 10% points in the community areas, the risk of developing depressive symptoms in the community’s older people as a whole would decrease by 10% and the risk of developing depressive symptoms in the 70–74 year age group could be almost the same or greater than that in the 65–69 year age group, which could be equivalent to more than five years of mental health rejuvenation. 

The strength of the present study is that it analyzed a large cohort sample of older people aged 65 years and older from 24 municipalities across Japan. In addition, we conducted a multilevel analysis of participation in volunteer groups and revealed the association with individual health; that is, the contextual effects considering the community characteristics. It is also important to note that the contextual effect was confirmed by the proportion of volunteer activities more than 1 day/month in the current study. This result supports previous studies reporting that volunteer participation at least 1 day/month is good for health [28]. Volunteer group participation 1 day/month is feasible and could be a good implication for public health experts to prevent depression among older people.

However, several limitations of this study should be noted. First, the response and follow-up rate were 67.1% and 59.8%, respectively. Therefore, there is a possibility of selection bias because those dropouts during the follow-up were associated with required long-term care certification or more serious conditions. Second, the questionnaire used in the present study did not ask participants the type of volunteer activities undertaken. Previous reports showed that Japanese older people tend to participate in volunteer activities such as academic, sports, culture and art promotion; community development; disaster prevention and crime prevention; long-term care prevention; parenting and health promotion [44]. Future studies should ask the participants the type of volunteer activities undertaken and investigate the community-level effect on individual health. Finally, depressive symptoms were not a clinical diagnosis but measured by a scale. However, a high sensitivity and specificity have been shown in previous studies [23] and are not likely to bias the results.

## 5. Conclusions

Older people living in community areas with a higher prevalence of volunteer group participation of older people are less likely to have subsequent depressive symptoms by 10% compared with those living in a community area with a lower prevalence of volunteer group participation. In other words, the study showed potential contextual effects in the prevention of developing depressive symptoms in older people. Promoting an increased number of older people participating in volunteer groups may be effective as a population approach to lower the risk of developing depressive symptoms among the older people living in community areas.

## Figures and Tables

**Figure 1 ijerph-18-07502-f001:**
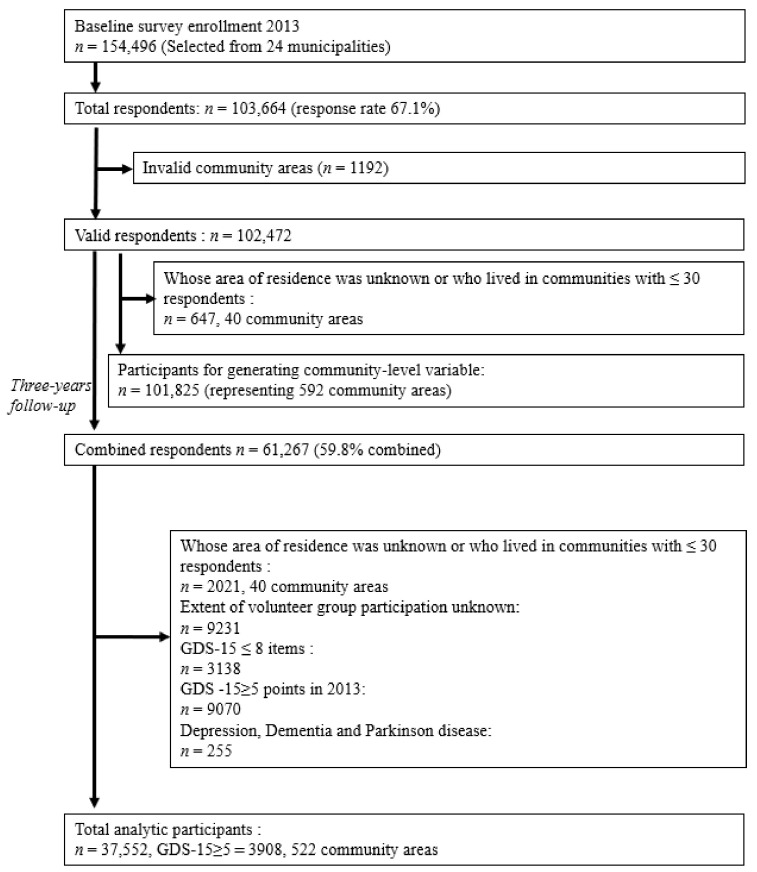
Flow of participants in the cohort study.

**Table 1 ijerph-18-07502-t001:** Descriptive statistics of individual-level covariates and incidence rates of depressive symptoms.

		Developing Depressive Symptoms (GDS-15 ≥ 5) ^a^	
Individual-Level Variables	Total N	Number of Cases	Cumulative Incidence
Total	37,552	3908	10.40%
Frequency of participation in a volunteer group			
≥4 day/week	509	36	7.10%
2–3 day/week	977	74	7.60%
1 day/week	1144	81	7.10%
1–3 day/month	3200	248	7.80%
A few times a day/year	3598	345	9.60%
Zero	28,124	3124	11.10%
Sex			
Man	17,575	1788	10.20%
Woman	19,977	2120	10.60%
Age, year			
65–69	12,346	1063	8.60%
70–74	12,879	1241	9.60%
75–79	7683	881	11.50%
80–84	3489	511	14.60%
≥85	1155	212	18.40%
Education, year			
<6	263	51	19.40%
6–9	13,165	1673	12.70%
10–12	14,849	1454	9.80%
≥13	8801	668	7.60%
Other and missing	474	62	13.10%
Annual equivalent income, JPY			
<2,000,000	13,905	1750	12.60%
2,000,000–3,999,999	13,925	1214	8.70%
≥4,000,000	4284	272	6.30%
Missing	5438	672	12.40%
Disease status in treatment			
Stroke no	34,413	3624	10.50%
Stroke yes	868	112	12.90%
Stroke missing	2271	172	7.60%
Hypertension no	19,354	2005	10.40%
Hypertension yes	15,927	1731	10.90%
Hypertension missing	2271	172	7.60%
Diabetes no	30,916	3225	10.40%
Diabetes yes	4365	511	11.70%
Diabetes missing	2271	172	7.60%
Ear disease no	33,451	3470	10.40%
Ear disease yes	1830	266	14.50%
Ear disease missing	2271	172	7.60%
Living alone			
No	32,829	3327	10.10%
Yes	3383	405	12.00%
Missing	1340	176	13.10%
Drinking status			
None	21,690	2397	11.10%
Past	1442	207	14.40%
Current	14,029	1261	9.00%
Missing	391	43	11.00%
Smoking status			
None	27,987	2863	10.20%
Past	5909	611	10.30%
Current	3246	382	11.80%
Missing	410	52	12.70%
BMI ^b^ (kg/m^2^)			
<18.5	2127	279	13.10%
≥18.5 and <25	26,106	2593	9.90%
≥25 and <30	7441	784	10.50%
≥30 and <50	679	73	10.80%
Missing	1199	179	14.90%

^a^ GDS-15: Geriatric Depression Scale 15. ^b^ BMI: body mass index.

**Table 2 ijerph-18-07502-t002:** Association of depressive symptoms with community-level and individual-level volunteer participation by a multilevel Poisson regression.

Model 1: Adjusted for Outcome, Individual-Level, Community-Level, Cross-Level, Age and Sex		
Total, N = 37,552	IRR ^b^	95% CI ^c^
Fixed effects		
Volunteer participation at the individual level	0.70	(0.63–0.78)
Proportion of volunteer participation at the community level 10% estimation ^a^	0.86	(0.77–0.97)
Cross-level interaction	1.06	(0.79–1.42)
Age, y		
65–69	1.00	
70–74	1.13	(1.04–1.22)
75–79	1.35	(1.23–1.47)
80–84	1.71	(1.54–1.90)
≧85	2.12	(1.83–2.46)
Sex		
Man	1.00	
Woman	1.04	(0.98–1.11)
Intercept (SE) ^d^	0.08	1.807
Random effect		
Ωμ (SE) ^d^	0.005	0.022
^a^. Cross-level interaction: community-level exposure 10% estimation × individual-level exposure.		
^b^. IRR: incidence rate ratio.		
^c^. CI: confidence interval.		
^d^. SE: standard error.		
**Model 2: Model 1 Adjusted for Community-Level Covariates**		
**Total, N = 37,552**	**IRR ^b^**	**95% CI ^c^**
Fixed effects		
Volunteer participation at the individual level	0.70	(0.63–0.77)
Proportion of volunteer participation at the community level 10% estimation ^a^	0.87	(0.78–0.98)
Cross-level interaction	1.06	(0.79–1.43)
Population density, persons per kilometer squared of inhabitable area		
Highest quartile (≥11,385)	1.00	
Second quartile (9028–11,378)	1.00	(0.86–1.16)
Third quartile (5194–8977)	1.06	(0.91–1.22)
Lowest quartile (<5158)	1.21	(1.06–1.38)
Total annual sunshine hours		
Highest quartile (≥ 2076)		
Second quartile (2047–2075)	0.92	(0.81–1.03)
Third quartile (1911–2047)	0.97	(0.83–1.13)
Lowest quartile (<1911)	0.98	(0.87–1.11)
Annual Rainfall (mm)		
Highest quartile (≥1622)	1.00	
Second quartile (1521–1621)	0.90	(0.76–1.06)
Third quartile (1483–1520)	0.98	(0.84–1.14)
Lowest quartile (<1483)	0.82	(0.72–0.95)
Age, y		
65–69	1.00	
70–74	1.13	(1.04–1.22)
75–79	1.34	(1.22–1.47)
80–84	1.70	(1.53–1.89)
≧85	2.09	(1.80–2.43)
Sex		
Man	1.00	
Woman	1.04	(0.98–1.11)
Intercept (SE) ^d^	0.08	0.008
Random effect		
Community-level variance		
Ωμ (SE) ^d^	0.036	0.008
^a^. Cross-level interaction: community-level exposure 10% estimation × individual-level exposure.		
^b^. IRR: incidence rate ratio.		
^c^. CI: confidence interval.		
^d^. SE: standard error.		
**Model 3: Model 2 Adjusted for Individual-Level Covariates**		
**Total, N = 37,552**	**IRR ^b^**	**95% CI ^c^**
Fixed effects		
Volunteer participation at the individual level	0.72	(0.65–0.80)
Proportion of volunteer participation at the community level 10% estimation ^a^	0.90	(0.80–1.01)
Cross-level interaction	1.06	(0.79–1.42)
Population density, persons per kilometer squared of inhabitable area		
Highest quartile (≥11,385)	1.00	
Second quartile (9028–11,378)	0.99	(0.85–1.15)
Third quartile (5194–8977)	1.02	(0.88–1.19)
Lowest quartile (<5158)	1.12	(0.99–1.28)
Total annual sunshine hours		
Highest quartile (≥2076)	1.00	
Second quartile (2047–2075)	0.91	(0.81–1.02)
Third quartile (1911–2047)	0.96	(0.83–1.12)
Lowest quartile (<1911)	0.96	(0.84–1.08)
Annual Rainfall (mm)		
Highest quartile (≥1622)	1.00	
Second quartile (1521–1621)	0.88	(0.75–1.04)
Third quartile (1483–1520)	0.98	(0.84–1.14)
Lowest quartile (<1483)	0.83	(0.73–0.96)
Age, y		
65–69	1.00	
70–74	1.08	(1.00–1.18)
75–79	1.25	(1.14–1.37)
80–84	1.55	(1.39–1.73)
≧ 85	1.88	(1.62–2.20)
Sex		
Man	1.00	
Woman	1.02	(0.94–1.10)
Education, y		
≧6	1.00	
<6	1.24	(0.94–1.64)
Other and missing	1.16	(0.85–1.58)
Annual equivalent income, JPY		
≧4,000,000	1.00	
<2,000,000	1.83	(1.61–2.09)
2,000,000-3,999,999	1.35	(1.19–1.54)
Missing	1.72	(1.50–1.99)
Disease status in treatment		
Stroke no	1.00	
Stroke yes	1.13	(0.93–1.36)
Hypertension no	1.00	
Hypertension yes	1.04	(0.98–1.11)
Diabetes no	1.00	
Diabetes yes	1.14	(1.04–1.30)
Ear disease no	1.00	
Ear disease yes	1.32	(1.16–1.50)
Family structure		
No	1.00	
Yes (living alone)	1.07	(0.97–1.20)
Missing	1.13	(0.97–1.32)
Drinking status		
None	1.00	
Past	1.21	(1.04–1.41)
Current	0.86	(0.79–0.93)
Missing	0.78	(0.51–1.18)
Smoking status		
None	1.00	
Past	1.10	(0.99–1.22)
Current	1.29	(1.15–1.44)
Missing	1.30	(0.89–1.90)
BMI ^e^ (kg/m^2^)		
≧18.5	1.00	
<18.5	1.24	(1.10–1.41)
Missing	1.24	(1.06–1.44)
Intercept (SE) ^e^	0.06	0.007
Random effect		
Community-level variance		
Ωμ (SE) ^e^	0.035	0.008

^a^. Cross-level interaction: community-level exposure 10% estimation × individual-level exposure. ^b^. IRR: incidence rate ratio. ^c^. CI: confidence interval. ^e^. BMI: body mass index. SE: standard error.

## Data Availability

Restrictions apply to the availability of these data. Data was obtained from Japan Gerontological Evaluation Study.

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
