# Peer review of "Community-Level Participation in Volunteer Groups and Individual Depressive Symptoms in Japanese Older People: A Three-Year Longitudinal Multilevel Analysis Using JAGES Data"

_ijerph, 2021, doi:10.3390/ijerph18147502_

Round 1

Reviewer 1 Report

Thank you for the opportunity to revise the manuscript entitled “ Community-Level Participation of Volunteer Groups and Individual Depressive Symptoms in Japanese Older People: A 3-Year Longitudinal Multilevel Analysis Using JAGES Data”. This is a study that investigates the associations of volunteer group participation on the development of depressive symptoms in older people in a large cohort of older adults from multiple (24) municipalities across Japan.

COMMENTS:

The manuscript does need to be reviewed for English grammar. 

Points and commas are used to separate decimal and thousands, respectively. But in some occasions commas are not used: i.e. Line 102 (1192), 170 (“(<5066.242, 5092.20-8929.175, 8967.723-11281.48, and ≥11300.39”) etc. Please check the whole manuscript, including tables.

Abstract:

"We used the 18 data of 154,496 people aged 65 years and older…” please indicate that the final N was 37,552.

Abbreviations are not necessary (JAGES, GDS-15, IRR, IC) since they are mentioned only once.

1. Introduction:

In general terms, there is too much information regarding the concepts of social capital, social contagion, collective efficacy, etc.  Which are interesting, but can distract the reader from the exact variables analyzed.

Line 53-54 : “And previous studies have reported that…” Please provide more than one reference (9).

Last paragraph: please add hypothesis.

2. Material and methods

Line 129: please define or provide more information regarding “volunteer group participation”.

Line 151: define drinking

BMI: please explain how BMI was obtained, add units (kg/m2) throughout the manuscript, and add a reference regarding the cutoff point 18.5.

3. Results

The dependent variable refers to the onset or development of depressive symptoms, but for instance, in line 225: “The variables which significantly correlated with depressive symptoms were…”. Please check this in all the results/discussion sections.

Overall, the results of the tables should be explained in a more detailed fashion, not only pointing out the significant results but also the “sense” of the associations… i.e.  the development of depressive symptoms is associated with greater age or BMI (or more or less than 18.5…) ? Or in categorical variables such as sex (male or female?), education, etc.

Check table footnotes regarding the description of the abbreviations: (superscripts a, b, etc) are used but do not appear in the tables.

4. Discussion

As mentioned above, I think that the results of the study are barely discussed. In turn, there is too much information regarding concepts such as collective efficacy or social contagion (the second and third paragraphs of the discussion).

As a suggestion, this section could start with one sentence that describes the main objective of this study.

Author Response

Dear. Reviewer 1

Thank you very much for your positive comments about this manuscript. 
We also received a lot of suggestions, which made our manuscript more comprehensible. 
Thank you for your suggestions.

Please see the attached file for our response to your suggestion.

Best regards,

Reviewer 2 Report

Article with a well contextualized introduction.

The tables provide a lot of information. The tables are too extensive considering the objectives of the study. The tables should be summarized or show the most important variables.

It would have been very interesting to know what activities are being carried out. Future research would broaden the scope of this study.

In the discussion section:

It would be necessary to compare the results obtained with similar studies. The discussion follows the line of the introduction, continuing to contextualize the main topic.

Author Response

Dear. Reviewer 2

Thank you very much for your positive comments about this manuscript. 
We also received a lot of suggestions, which made our manuscript more comprehensible. 
Thank you for your suggestions.

Please see the attached file for our response to your suggestion.

Best regards,

Reviewer 3 Report

It is exciting material in the face of the need to investigate the contextual effect on depressive symptoms in older people. But to examine it, it is mandatory to develop theoretical models of the event causation that extend across levels and explain how group-level and individual-level variables interact in shaping the topic of interest, as discussed by Diez-Roux (1998). Considering that, I would recommend that:

- It is necessary to consider the time frame of the independent variable “subsequent depressive symptoms”;

- to present the theoretical models of disease causation that extend across levels – including individual and macro-level variables included in the studies.

I take the opportunity to recommend a review considering classical epidemiological papers that discuss context analysis, as shown below.

References:

DIEZ-ROUX, Ana V. Bringing context back into epidemiology: variables and fallacies in multilevel analysis. American journal of public health, v. 88, n. 2, p. 216-222, 1998.

MERLO, Juan et al. A brief conceptual tutorial of multilevel analysis in social epidemiology: using measures of clustering in multilevel logistic regression to investigate contextual phenomena. Journal of Epidemiology & Community Health, v. 60, n. 4, p. 290-297, 2006.

Author Response

Dear. Reviewer 3

Thank you very much for your positive comments about this manuscript. 
We also received a lot of suggestions, which made our manuscript more comprehensible. 
Thank you for your suggestions.

Please see the attached file for our response to your suggestion.

Best regards,

Round 2

Reviewer 1 Report

The authors have addressed the comments and suggestions I had on the initial manuscript, I think this revised version is an improvement and the paper certainly reads better now.

Reviewer 3 Report

I have no comments.